METHODS AND RESOURCES

# Attenuation hotspots in neurotropic human astroviruses

**Hashim Ali**[1], **Aleksei Lulla**[2], **Alex S. Nicholson**[3], **Jacqueline Hankinson**[1], **Elizabeth B. Wignall-Fleming**[1], **Rhian L. O'Connor**[1], **Diem-Lan Vu**[4], **Stephen C. Graham**[1], **Janet E. Deane**[3], **Susana Guix**[4], **Valeria Lulla**[1]*

1 Department of Pathology, University of Cambridge, Addenbrookes Hospital, Cambridge, United Kingdom, 2 Department of Biochemistry, University of Cambridge, Cambridge, United Kingdom, 3 Cambridge Institute for Medical Research, University of Cambridge, Cambridge, United Kingdom, 4 Enteric Virus Group, Department of Genetics, Microbiology and Statistics, Research Institute of Nutrition and Food Safety (INSA-UB), University of Barcelona, Barcelona, Spain

* vl284@cam.ac.uk

## Abstract

During the last decade, the detection of neurotropic astroviruses has increased dramatically. The MLB genogroup of astroviruses represents a genetically distinct group of zoonotic astroviruses associated with gastroenteritis and severe neurological complications in young children, the immunocompromised, and the elderly. Using different virus evolution approaches, we identified dispensable regions in the 3′ end of the capsid-coding region responsible for attenuation of MLB astroviruses in susceptible cell lines. To create recombinant viruses with identified deletions, MLB reverse genetics (RG) and replicon systems were developed. Recombinant truncated MLB viruses resulted in imbalanced RNA synthesis and strong attenuation in iPSC-derived neuronal cultures confirming the location of neurotropism determinants. This approach can be used for the development of vaccine candidates using attenuated astroviruses that infect humans, livestock animals, and poultry.

**Data Availability Statement:** All relevant data are within the paper and its Supporting Information files. Quantitative data can be found in the spreadsheet S1 Data. A separate tab is associated with each Figure and Supporting Information.

## Introduction

Human astroviruses (HAstVs) belong to the genus *Mamastrovirus*, family *Astroviridae* and are a common cause of gastroenteritis in children, the elderly, and immunocompromised adults [1]. Lately, the HAstV group of the *Astroviridae* family has expanded to include new groups of viruses unrelated to the 8 previously described classic HAstV serotypes (Fig 1A). These new human astrovirus groups are more closely related to certain animal astroviruses than to the classical HAstVs, suggesting zoonotic transmission [1]. One of these groups is named MLB, after the first novel human astrovirus described in 2008 in Melbourne (Australia) identified in feces of pediatric patients with gastroenteritis. Later, the MLB group of HAstVs was assigned to a neurovirulent group of astroviruses due to the association with severe cases of meningitis/encephalitis, febrile illness, and respiratory syndromes [2–4]. Interestingly, it was recently shown that astroviruses found in the fecal samples of macaque monkeys were genetically similar to human astrovirus MLB and caused chronic diarrhea [5].

Uncropped images are found in S1 Raw Images.
GenBank accession numbers: ON398705,
ON398706.

**Funding:** This work was funded by a Sir Henry Dale
Fellowship (220620/Z/20/Z) from the Wellcome
Trust and the Royal Society, an Isaac Newton
Trust/Wellcome Trust ISSF/University of
Cambridge Joint Research Grant and MRC project
grant (MR/T000376/1) to V.L. J.E.D and A.S.N. are
supported by a Wellcome Trust Senior Research
Fellowship (219447/Z/19/Z) awarded to J.E.D. R.L.
O. is supported by the MRC DTP Studentship
https://www.ukri.org/councils/mrc/ https://
wellcome.org/ https://www.research-strategy.
admin.cam.ac.uk/research-funding/internal-
funding-opportunities/institutional-sponsorship-
grants/wellcome-trust The funders had no role in
study design, data collection and analysis, decision
to publish, or preparation of the manuscript.

**Competing interests:** The authors have declared
that no competing interests exist.

**Abbreviations:** CP, capsid polyprotein; CPE,
cytopathic effect; HAstV, human astrovirus; hpi,
hours post infection; IU, infectious unit; iPSC,
induced pluripotent stem cell; MOI, multiplicity of
infection; ORF, open reading frame; PFA,
paraformaldehyde; RdRp, RNA-dependent RNA
polymerase; RG, reverse genetics; SG,
subgenomic; UTR, untranslated region; wt, wild-
type.

HAstVs are small, non-enveloped, icosahedral viruses with positive-sense single-stranded RNA genome containing 5′ untranslated region (UTR), 4 open reading frames (ORF1a, ORF1b, ORFX, and ORF2), and a 3′ UTR with poly A tail [6,7]. ORF1a encodes nonstructural polyprotein nsP1a, ORF1b is expressed via ribosomal frameshifting mechanism and encodes the RNA-dependent RNA polymerase (RdRp). The subgenomic (SG) RNA encodes 2 ORFs–ORF2 and ORFX, the latter encoding a viroporin [7]. The product of the ORF2 coding sequence is translated into the structural capsid polyprotein (CP) of about 72 to 90 kDa, depending on the virus strain [8], which then undergoes C-terminal cleavage by cellular caspases [9]. Despite the required function of caspases, the astrovirus release is described as an unclassified nonlytic process [10]. In some astroviruses, the structural polyprotein is cleaved by trypsin resulting in the formation of truncated (25 to 34 kDa) proteins. Maturation of the astrovirus capsid proteins is a very dynamic process, transforming the virus from a noninfectious intracellular form (VP90) to a primed extracellular form (VP70) and finally generating an icosahedral infectious mature virion (VP34/27/25). However, some concerns remain to be addressed to fully understand the astroviruses capsid assembly and maturation. In particular, what differences in the capsid protein of the MLB genotypes make them different from the classical astroviruses and what determines their infectivity? It has been shown that trypsin treatment of classical astroviruses increases their infectivity [11,12], while not affecting the MLB genotypes [2], further confirmed by recently revealed differences in HAstV and MLB capsid structure [13]. The mechanism of CP cleavage and the functional role of cleaved CP in the MLB group of astroviruses are not yet understood. It is also unclear if MLB astroviruses exploit cellular proteases other than trypsin to process the CP and how this impacts the infectivity of virus particles.

Growing evidence suggests that astroviruses are found globally, infecting a wide range of species, and have the potential for recombination, rapid evolution, and can adapt to different hosts [5,14–19]. Unfortunately, many astrovirus groups have remained overlooked for decades because of the absence of molecular tools, such as infectious clones and replicons. Therefore, developing a robust reverse genetics (RG) system for the nonclassical human astroviruses is essential to understand the basic biology, evolution, and host–virus interplay.

In 1997, Matsui's group established the first RG system for the human astrovirus serotype 1 to rescue infectious viral particles [20]. This system has been successfully used and has shed light on multiple aspects of astrovirus replication and pathogenesis. HAstV1 RG system requires 2 cell lines to recover infectious particles: the transfection of BHK-21 cells with *in vitro* transcribed viral RNA and then propagation of the obtained supernatant in the permissive Caco2 cells in the presence of trypsin. Several DNA-based RG systems were developed, including efficient chimeric HAstV1/8 RG system [21,22]; however, all of them relied on 2 cell lines and were limited to classical human astroviruses. So far, RG systems for 2 nonhuman astroviruses were developed: first for the avian astrovirus by using duck astrovirus (DAstV) genome of D51 strain [23] and second for porcine astrovirus (PAstV1-GX1) [24]. Although both nonhuman RG systems allow the recovery of infectious viral particles, these systems also rely on the 2 cell lines.

It is therefore essential to develop the RG system for neurotropic astroviruses to understand the molecular determinants for neurotropism and neurovirulence. Here, we report the RG system for 2 nonclassical human neurotropic astroviruses that relies on a single cell line and can be used to rescue and propagate MLB1 and MLB2 human astroviruses. We also developed a set of detection tools as well as replicon systems for both MLB astroviruses. Using this system, we identified and characterized attenuation hotspots located at the 3′ end of the MLB genomes that impact neurovirulence of these viruses. In the future, this RG system will deepen the

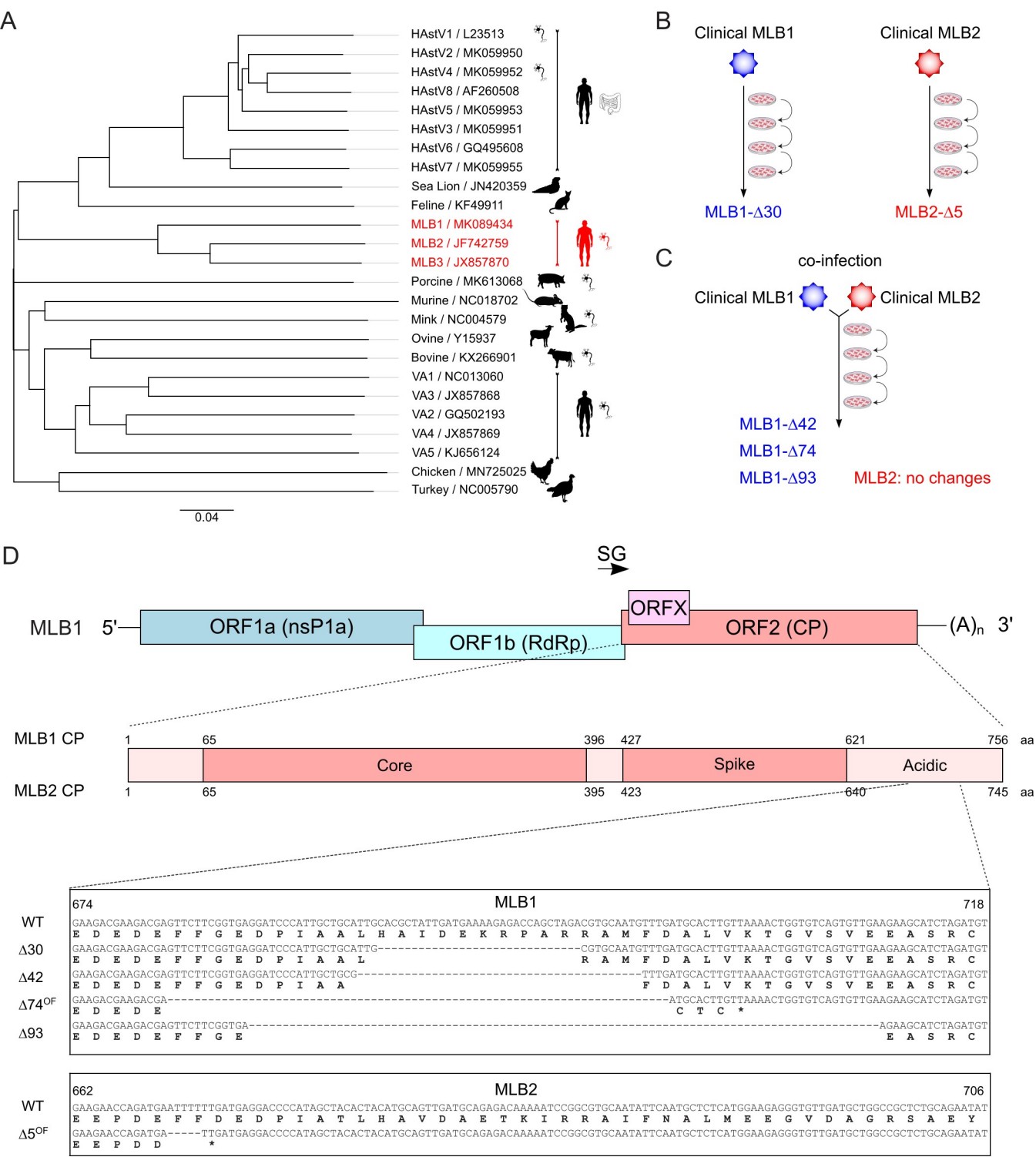

**Fig 1. Classification and attenuation of MLB astroviruses. (A)** Simplified phylogenetic tree for the *Astrovirus* genus. The tree is based on full nucleotide sequences available for indicated species. The pictogram of the intestine or neuron indicates the tropism associated with astrovirus strains. Some astrovirus genotypes labeled with the neuron icon are associated with neuropathology. Neurotropic MLB strains are shown in red. **(B)** The evolution experiment was performed for MLB1 and MLB2 astroviruses. **(C)** The coevolution experiment was performed for MLB1 and MLB2 astroviruses. **(D)** Nucleotide and amino acid sequences of MLB1 and MLB2 viruses containing deletions identified in evolved MLB virus stocks.

understanding of molecular virology of MLB-group astroviruses and allow the design of tools to address open questions on viral evolution, replication, packaging, and pathogenesis.

## Results

### Evolution and cell culture adaptation of neurotropic MLB astroviruses

The attenuation of viruses by serial cultivation *in vitro* or in abnormal hosts dates back to the 19th century [25]. Passaging pathogenic viruses in chicken embryos, mice and/or cell cultures led to the development of several vaccine candidates including polio, measles, yellow fever, rubella, and many other pathogens [26,27]. Therefore, we hypothesized that this strategy could be used to attenuate MLB astroviruses. Clinical MLB isolates were obtained from stool (MLB1) and cerebrospinal fluid (MLB2) of infected patients and passaged in susceptible cell lines, as previously described [2] (Fig 1B). Sequencing of a passaged clinical MLB1 isolate revealed a deletion of 30 nucleotides in the 3′ end of the genome spanning into the coding sequence of CP. A similar region was affected in the passaged MLB2 clinical isolate—a single out-of-frame deletion of 5 nucleotides in the 3′ part of the genome (Fig 1D).

Another strategy for directed virus evolution is based on coinfection of closely related virus species [28]. The better replicating "partner" can either out-compete or complement the replication of another virus. Complementation would result in faster evolution of less "fit" viruses. To test this hypothesis, we coinfected Huh7.5.1 cells at a multiplicity of infection (MOI) 0.1 with MLB1 and MLB2 viruses (Fig 1C). This strategy can facilitate complementation after the first cycle of virus replication while avoiding the accumulation of defective interfering genomes common for the high MOI [29]. This resulted in simultaneous replication and propagation of both strains on passaging without detected out-competition or recombination for 10 consecutive passages. Interestingly, no changes were observed in MLB2 genomes; however, a mixture of several in-frame and out-of-frame deletions was detected in the 3′ part of the MLB1 genome further confirming the instability of the 3′ region in this virus (Fig 1D).

Elucidating the functional significance of the identified deletions requires MLB astrovirus detection tools and would be dramatically accelerated by the establishment of a robust RG system. We, therefore, aimed to create these essential tools.

### Cell culture models and detection tools for neurotropic MLB1 and MLB2 astroviruses

First, we developed a set of essential tools for specific immune detection of virus infection. The folded region of MLB1 capsid protein corresponding to amino acids 61–396 of ORF2-encoded polyprotein possessing a C-terminal 8×His-tag (Fig 2A) was used for bacterial expression and affinity purification, resulting in homogeneous $CP_{NTD}$ protein (Fig 2B). The purified recombinant protein was used for the production of highly sensitive antibodies allowing the detection of ≤1 ng of the purified $CP_{NTD}$ of MLB1 (Fig 2C). Due to 95% identity between corresponding domains of MLB1 and MLB2 CPs, polyclonal antibodies were expected to cross-detect capsid proteins derived from both strains. Indeed, it specifically recognized capsid proteins from MLB1- and MLB2-infected cells (Fig 2C).

When passaging clinical isolates, we noticed that the Huh7.5.1 cell line supports MLB replication and results in a moderate cytopathic effect (CPE). MLB clade viruses were previously reported to replicate in Huh7 and Huh7.5 cell lines with the ability to establish a persistent infection on passaging [2]. Presumably, in contrast to the immune-competent Huh7 cells, the more susceptible Huh7.5.1 cell line [7] can allow for enhanced replication and development of CPE. This cell line was also reported to support active replication of the classical human

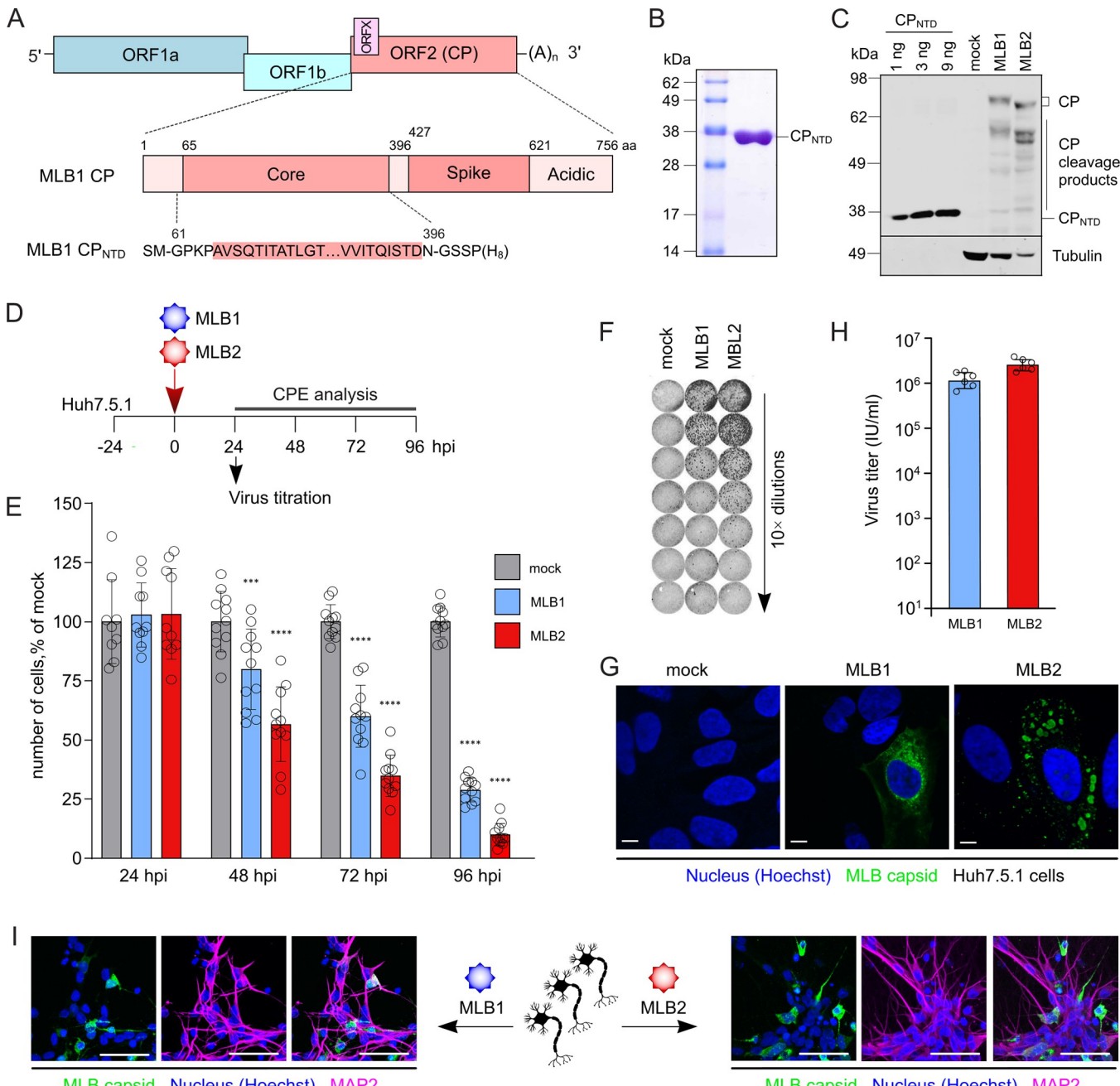

**Fig 2. Purification of CP_NTD and immunodetection of MLB1- and MLB2-infected Huh7.5.1 cells and iPSC-derived neurons.** **(A)** Schematic representation of MLB1 genome and location of CP_NTD. Lower panel represents the sequence of recombinant CP_NTD. ORF, open reading frame; CP, capsid polyprotein; NTD, N-terminal domain. **(B)** Coomassie-stained SDS-PAGE profile of the purified CP_NTD from *E. Coli*. **(C)** Huh7.5.1 cells were infected with MLB1 and MLB2 viruses at MOI 0.1 and incubated for 48 h. CP was detected using the antibody generated against CP_NTD. Purified CP_NTD was used for detection limit assessment (1–9 ng). **(D)** Experimental setup to determine the CPE and titration for MLB1 and MLB2 infection. **(E)** Huh7.5.1 cells were infected at an MOI 1 and incubated for indicated periods, then stained and imaged. Hoechst-stained nuclei were counted from 12 images (approximately 200 cells per image) and normalized to mock-infected samples. Data are mean ± SD. ***$p < 0.001$, ****$p < 0.0001$ using two-way ANOVA test against mock. **(F)** Huh7.5.1 cells were seeded on 96-well plate and infected with 10-fold serial dilutions of MLB1 and MLB2 astroviruses, fixed at 20 hpi, permeabilized, stained with anti-CP antibody, and imaged by LI-COR. **(G)** Huh7.5.1 cells were infected with MLB1 and MLB2 viruses and incubated for 24 h. Representative confocal images of fixed and permeabilized cells visualized for CP (green) and stained for nuclei (Hoechst, blue) are shown. Scale bars are 10 μm. **(H)** Huh7.5.1 cells were infected with MLB1 and MLB2 virus stocks at MOI 0.1 and incubated for 72–120 h. Virus titers were determined from 6 independent experiments. Data are mean ± SD. **(I)** Partially differentiated i3Neurons were seeded on IBIDI plates, differentiated into mature glutamatergic neurons, infected with MLB1 and MLB2 viruses, and incubated for 48 (MLB2) or 96 (MLB1) h. Representative confocal images of fixed and permeabilized cells visualized for MLB CP (green), neuronal marker MAP2 (magenta) and stained for nuclei (Hoechst, blue) are shown. Scale bars are 50 μm. All uncropped images can be found in the

Supporting information file as S1 Raw Images. All individual quantitative observations that underlie the data can be found in S1 Data file. CPE, cytopathic effect; hpi, hours post infection; MOI, multiplicity of infection.

astrovirus 1 (HAstV1) [30]. The CPE was apparent for MLB2 at 24 to 48 h post infection (hpi), whereas slower replicating MLB1 showed CPE at 48 to 72 hpi, reaching >60% cell death at 3 days post infection for MLB2 and at 4 days post infection for MLB1 (Fig 2D and 2E). The differences in CPE were also confirmed in the independent automated cytotoxicity screening using live-cell imaging (S1 Fig).

The permissiveness of the Huh7.5.1 cell line allowed the development of a virus titration system. Cells cultured on 96-well plates were infected with 10-fold dilutions of MLB1 and MLB2 stocks, fixed and stained with $CP_{NTD}$-antibody using in-cell near-infrared fluorescence-based detection (Fig 2D and 2F). The cytoplasmic distribution of CP was confirmed by confocal microscopy further demonstrating that both MLB1 and MLB2 can be detected 24 hpi (Fig 2G). Since no released virus could be detected after 20 hpi, this timepoint was utilized for single-round infection experiments like titration of the virus stocks. Efficient virus release was detected at 48 to 72 hpi for MLB2 and at 96 to 120 hpi for MLB1, reaching $1–3 \times 10^6$ infectious units (IU) per ml (Fig 2H).

Finally, to confirm the neurotropic properties of MLB astroviruses, we developed a physiologically relevant system to infect and monitor MLB infection in neurons. The neurotropism of MLB astroviruses was previously described and indicates their ability to infect cells of neuronal origin [2,4]. The recently developed methodology to efficiently differentiate human-induced pluripotent stem cells (iPSCs) into isogenic cortical glutamatergic neurons ($i^3$Neurons) [31] provided a suitable platform to experimentally assess neurotropic properties of MLB1 and MLB2 viruses. Both viruses resulted in efficient infection at 48 hpi (MLB2) and 96 hpi (MLB1) further confirming the ability of these viruses to infect postmitotic neuronal cells (Fig 2I).

## Development, annotation, and assessment of infectious clones for MLB1 and MLB2 viruses

The 5′ and 3′ terminal consensus sequences were used to design specific primers to amplify MLB1 (MK089434) and MLB2 (JF742759) full-length genomes. The entire genomes of MLB1 (Fig 3A) and MLB2 (Fig 3B) were cloned into the T7 promoter-containing plasmid using a single-step ligation-independent cloning. The obtained plasmids were sequenced and the ORFs and functional elements were annotated based on homology with other astroviruses (GenBank accession numbers: ON398705, ON398706). The resulting differences between recombinant and clinical isolate sequences have arisen due to polymorphisms present in initial clinical isolates. To produce recombinant viruses, the infectious clones of MLB1 and MLB2 were linearized and full genomic RNAs were synthesized *in vitro* using T7 RNA polymerase.

The ability of MLB astroviruses to replicate in Huh7.5.1 cells provides a great advantage for the development of the RG system: A single cell line can be used for both virus rescue and propagation—something that is not possible for other astroviruses with cell tropism restricted to non-transfectable cell lines [22]. Huh7.5.1 cells were electroporated with transcribed RNAs and incubated for 48 h (MLB2) or 72 h (MLB1) until the appearance of CPE. The supernatants were titrated and passaged at MOI 0.1 followed by titration and sequencing of resulting viral genomes (Fig 3C). The obtained recombinant viruses recapitulated the growth properties of the original clinically isolated MLB astroviruses [2], reaching final titers of $10^6$ to $10^7$ IU/ml. Consistent with previous findings [2], the increase of the virus in the extracellular fraction was

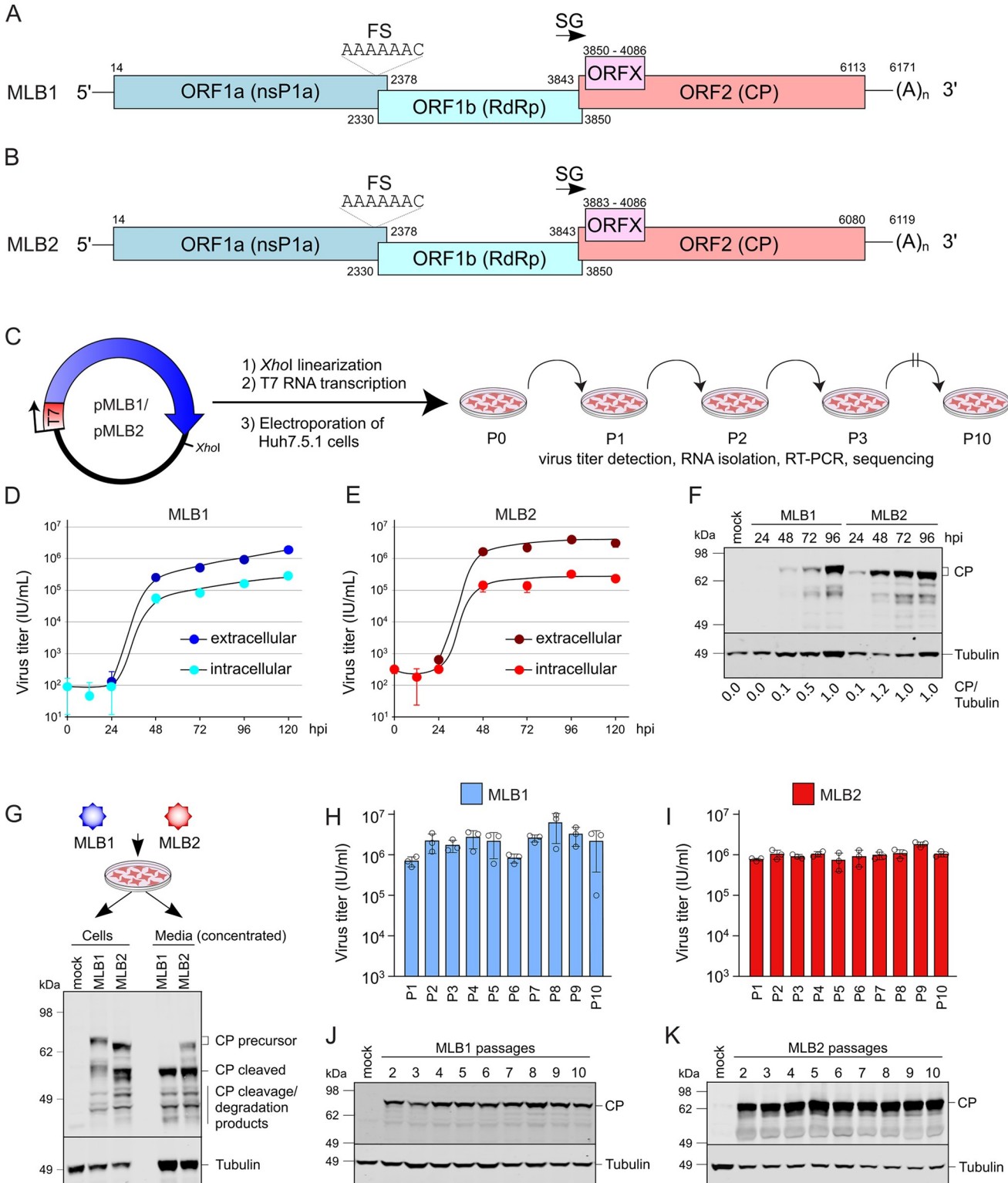

**Fig 3. Generation and validation of RG system for MLB1 and MLB2 astroviruses. (A, B)** Schematic representation of MLB1 (A) and MLB2 (B) genomes used to generate infectious clones. MLB genome elements: ORF, open reading frame; RdRp, RNA-dependent RNA polymerase; CP, capsid polyprotein; FS, frameshift site; SG, subgenomic promoter. **(C)** Strategy for a plasmid-derived RG system for MLB1 and MLB2. MLB cDNAs contain the entire genome flanked by the T7 promoter and *Xho*I linearization site. Huh7.5.1 cells were electroporated with full-genome T7 transcripts, the collected virus was used for serial passages in the same cell line. **(D, E)** Multistep growth curves of MLB1 (D) and MLB2 (E) on Huh7.5.1 cells (the second passage).

Cells were infected at an MOI 0.1, and virus titer was measured from the intracellular and extracellular fractions in triplicates. Data are mean ± SD. **(F)** Cells were infected at an MOI 0.1, harvested at 48 hpi, and analyzed by western blotting with anti-CP and anti-tubulin antibodies. **(G)** Huh7.5.1 cells were infected with MLB1 and MLB2 at an MOI 0.1 and incubated for 72 h. The cell- and media-derived samples were harvested and analyzed by western blotting. **(H, I)** Huh7.5.1 cells were infected in triplicates with MLB1 (H) and MLB2 (I) at an MOI 0.1 and incubated for 72–120 h until full CPE. Total virus titers of 10 serial passages were determined in triplicates (*n* = 2 independent experiments). Data are mean ± SD. **(J, K)** Analysis of CP expression in Huh7.5.1 cells infected with 2–10 passage of MLB1 (J) and MLB2 (K). Cells were infected at an MOI 0.1, harvested at 48 hpi, and analyzed by western blotting. All uncropped images can be found in the Supporting information file as S1 Raw Images. All individual quantitative observations that underlie the data can be found in S1 Data file. CPE, cytopathic effect; hpi, hours post infection; MOI, multiplicity of infection; RG, reverse genetics.

higher for MLB2 than for MLB1 (Fig 3D and 3E). The infection process coincided with the accumulation of capsid proteins, corresponding to observed increased virus production in these samples (Fig 3F). To get an estimation of the nature of smaller CP products, CP-specific products derived from cellular and media samples were analyzed. Interestingly, we predominantly observed cleaved CP form in the media-derived samples suggesting possible extracellular cleavage (Fig 3G) in analogy to classical HAstV strains [32]. Passaging of the MLB1 and MLB2 viruses resulted in similar virus titers (Fig 3H and 3I) and consistent production of capsid proteins (Fig 3J and 3K).

## Assessing genome stability of MLB1 and MLB2 viruses

To evaluate the MLB virus genome stabilities, we performed serial passaging by using RG-derived MLB1 and MLB2 recombinant viruses. The passaging was performed in biological duplicate, starting from *in vitro* RNA transcripts. No changes were detected in the passaged recombinant MLB2 virus, suggesting that its genome is stable in Huh7.5.1 cells. The slower replicating recombinant MLB1 was less stable and accumulated mutations in the 3′ part of the genome. An out-of-frame single-nucleotide insertion was detected at passage 3, that coexisted with wild-type (wt) MLB1 resulting in continuous coinfection for 7 consecutive passages (Table 1). A distinct cluster of mutations at the C-terminal end of CP was identified in the second experiment (Table 1) suggesting instability of RG MLB1 genomes during longer virus passaging.

To put these mutations and previously identified deletions (Fig 1D) in the context of naturally occurring changes in MLB genomes, the analysis of all available GenBank MLB astrovirus sequences was performed. As expected, the 3′ region of the MLB2 genome was very conserved: Few amino acid variations and no deletions were found in publicly available MLB2 genome sequences (Fig 4A). In contrast, the 3′ region of MLB1 was more diverse, containing multiple changes throughout the analyzed region as well as 1 amino acid deletion upstream of the experimentally observed deletion region (Fig 4B), consistent with the mutation- and deletion-prone nature of the 3′ region of MLB1 genome.

**Table 1. Genome changes in evolved recombinant MLB1 and MLB2 astroviruses.**

| Virus | Mutations (position in the genome) |
|---|---|
| Recombinant MLB1 | Experiment 1:<br> • insertion of A (6017) at passage 3, premature termination of CP at 729 aa, coexist with wt for the next 7 passages<br>Experiment 2:<br> • C5059T (P406L in CP) at passage 10<br>Cluster of mutations in CP: A6030G (K730E), AA6035GG (R732G), A6046G (Y735C) at passages 2–10 |
| Recombinant MLB2 | no changes detected |

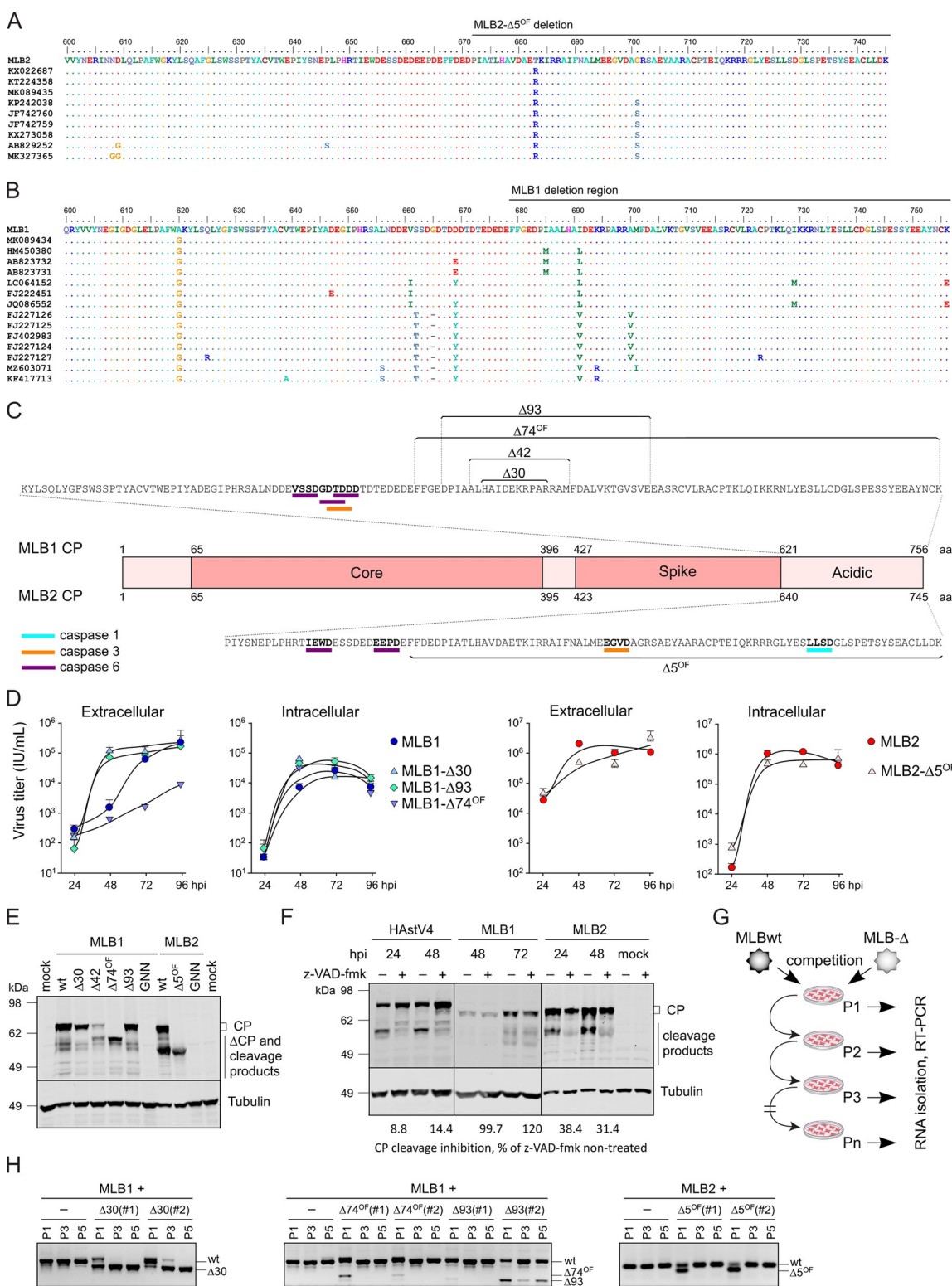

**Fig 4. Analysis of C-terminal part of MLB1 and MLB2 astrovirus CP.** (**A**) Analysis of publicly available sequences of C-terminal part of MLB2 CP. (**B**) Analysis of publicly available sequences of C-terminal part of MLB1 CP. The deletion region is indicated on top of MLB1 and MLB2 alignments (A, B). (**C**) Analysis of the C-terminal part of MLB1 and MLB2 CP for putative caspase cleavage sites using Procleave software [33]. The sequences in bold indicate caspase 1, 3, and 6 putative cleavage sites with a probability score of >0.7. The regions of MLB1 (top) and MLB2 (bottom) deletions are shown. (**D**) Huh7.5.1 cells were infected with indicated recombinant viruses of

MLB1 (left) and MLB2 (right) at an MOI 0.5 in triplicates, the virus was collected at indicated times post infection, and titer was measured for both extracellular and intracellular fractions. Data are mean ± SD. **(E)** Analysis of CP expression in Huh7.5.1 cells infected with second passage of MLB1 and MLB2 wt and mutant viruses (MOI 0.1). Cell lysates were harvested at 48 hpi and analyzed by western blotting with anti-CP and anti-tubulin antibodies. GNN is RdRp knock-out recombinant virus (GDD to GNN). **(F)** The effect of pan-caspase inhibitor z-VAD-fmk on CP processing during infection with classical human astrovirus 4 (HAstV4), MLB1 and MLB2 astroviruses. Caco2 cells were infected with HAstV4 and Huh7.5.1 cells were infected with MLB1 and MLB2 astroviruses at MOI 1 in the presence or absence of z-VAD-fmk. Cell lysates were harvested at indicated hpi and analyzed by western blotting with 8E7 antibody against HAstV CP (for HAstV4), anti-CP (for MLB), and anti-tubulin antibodies. The average inhibition of CP cleavage was quantified from 3 independent experiments. **(G)** The schematic representation of the virus competition experiment where Huh7.5.1 cells were coinfected with wt and mutant viruses and passaged. Virus RNA was isolated and used for RT-PCR to detect virus-specific fragments. **(H)** The RT-PCR fragments from the competition experiment were analyzed by agarose electrophoresis. The fragments corresponding to the expected size of each PCR product are shown on the right. All uncropped images can be found in the Supporting information file as S1 Raw Images. All individual quantitative observations that underlie the data can be found in S1 Data file. CP, capsid polyprotein; hpi, hours post infection; MOI, multiplicity of infection; wt, wild-type.

## Generation and characterization of recombinant MLB viruses with deletions

To investigate the individual roles of truncations identified in the passaging experiments (Fig 1D), we employed the RG system to create a set of recombinant viruses with the identified deletions (Fig 4C). The growth properties of recombinant viruses were analyzed for both intra- and extracellular virus production (Fig 4D). Compared to wt MLB1, both in-frame mutants (Δ30 and Δ93) showed faster virus release, whereas out-of-frame MLB1-Δ74$^{OF}$ showed delayed growth kinetics. Intracellular virus production was similar between MLB1 and mutant viruses. Another out-of-frame mutant, MLB1-Δ42$^{OF}$, grew to low titers ($\leq 10^4$ IU/ml) and displayed decreased protein production (Fig 4E) that precluded the use of this mutant in comparative experiments. MLB2-Δ5$^{OF}$ demonstrated a slight delay in virus growth, eventually replicating to wt levels for both extra- and intracellular viruses (Fig 4D). All mutant viruses demonstrated reduced cytotoxicity in infected cells (S1 Fig), supporting previous results that suggested a greater propensity to establish persistent infection [2].

The analysis of infected cells was performed using a CP$_{NTD}$-recognizing antibody, which was expected to recognize only N-terminal products of CP (Fig 4E). Similarly to related HAstVs, the C-terminal domain of MLB1 and MLB2 CP contains putative caspase cleavage sites, which could lead to the programmed C-terminal cleavage of CP. The predicted caspase cleavage sites were mapped for both MLB1 and MLB2 C-terminal domains of the CP (Fig 4C). Consistent with experimental observations, the cleaved product of MLB2 CP corresponds to the size of CP in MLB2-Δ5$^{OF}$ mutant suggesting caspase cleavage is taking place in this region (Fig 4C and 4E). The shorter forms of MLB1 deletion mutants seem to locate downstream of predicted caspase cleavage sites and potentially affect the C-terminal cleavage of CP in different ways: Δ30 and Δ93 follow the wt-like processing of CP, Δ74$^{OF}$ results in a shorter CP form, whereas Δ42 have both CP forms present. Notably, both Δ74$^{OF}$ and Δ42 lack the lower cleavage product, presumably one of the C-terminal truncated forms (Fig 4C and 4D). The integrity and stability of resulting mutant viruses over 3 passages were confirmed by RT-PCR and sequencing. To confirm the predicted caspase cleavage of CP, the Caco2 cells infected with classical human astrovirus 4 (HAstV4) and Huh7.5.1 cells infected with MLB1 and MLB2 were incubated in the absence or presence of pan-caspase inhibitor z-VAD-fmk. Consistent with the previously published results for classical HAstVs [9], both HAstV4 and MLB2 viruses resulted in the inhibition of CP cleavage in response to caspase inhibition (Fig 4F). In contrast, MLB1 was not sensitive to the inhibition of caspase-mediated processes (Fig 4F), suggesting that other cellular or viral proteases may be involved in the maturation of structural polyprotein of MLB1. These results support differences observed in the processing of CP that contains deletions in the C-terminal region.

The growth delay and altered CP processing in mutant viruses with deletions have raised the question of what selective advantage these mutations confer during passaging. The initial evolution experiments resulted in a mixture of defective genomes that could trans-complement each other but did not have a distinct "winner" except for MLB1-Δ30 which was the only mutant found in passaged clinical MLB1 (Fig 1B). Thus, we performed direct competition experiments between the wt and mutant MLB viruses (Fig 4G). Consistent with previous findings (Fig 1B) and virus growth experiments (Fig 4D), in-frame MLB1-Δ30 out-competed wt MLB1, thus confirming its advantage in Huh7.5.1 cells (Fig 4H, left panel). Another in-frame mutant (MLB1-Δ93) was out-competed either by other deletions that arose from unstable MLB1 or by coexisting with wt MLB1 during passaging (Fig 4H, middle panel). Out-of-frame deletion viruses (MLB1-Δ74$^{OF}$ and MLB2-Δ5$^{OF}$) were outcompeted by wt viruses (Fig 4H), confirming their slower replication properties in virus growth experiments (Fig 4D). Taken together, these results indicate a growth advantage for the smaller in-frame deletion mutant MLB1-Δ30 that was previously shown to solely out-compete wt MLB1 during the evolution of clinical MLB1 (Fig 1B).

## Analysis of RNA replication properties of truncated MLB viruses

The deletions that occurred in the 3′ end of the genome could potentially affect the replication properties of the virus, considering the unequivocal importance of 3′ UTR in the replication of (+)ssRNA viruses. Multiple structured RNA elements are predicted in the 3′ UTR of MLB1 and MLB2 genomes, including 3 stable stem-loops mapped to the MLB1 deletion region (Fig 5A). The 5-nucleotide deletion in MLB2 was not mapped to the structured RNA region, suggesting possible involvement of short- and/or long-range RNA interactions or other compensatory mechanisms including the processing of CP (Fig 4E). All 4 deletions in MLB1 were mapped to the predicted stem-loop RNA structures (Fig 5A) and could directly affect the RNA replication process.

To evaluate this hypothesis and assess the importance of overlapping 3′ RNA structures, replicon systems for MLB1 and MLB2 were created. In replicon systems, the RNA replication is evaluated via a fluorescent or luminescent reporter gene that replaces the structural proteins. The preserved RNA elements contain intact SG promoter, 5′ and 3′ UTRs, functional RdRp, and other components of the RNA replication machinery. This enables visualization and/or quantification of the SG reporter activity while avoiding the packaging step via deletion of the capsid region. Similar to a previously developed HAstV1-based replicon system [7], the genomes of MLB1 and MLB2 infectious clones were modified as demonstrated in Fig 5B. The 2A-cleavage-mediated Renilla luciferase (RLuc) or mCherry reporters were used to quantify and visualize the protein expression from the SG promoter. The transfection of both MLB1R-mCherry and MLB2R-mCherry replicons resulted in the expression of reporter mCherry (Fig 5C). The activity of MLB replicons expressing RLuc was monitored over time in 2 different cell lines (Fig 5D). Strong replication was detected for MLB2 replicon in Huh7.5.1 (1,251-fold) and HEK293T (568-fold) cells, when compared to replication-deficient but translation-competent GNN mutant replicon (with GDD catalytic RdRp motif changed to GNN). Lower replication levels were observed for MLB1 replicon in both Huh7.5.1 (379-fold) and HEK293T (100-fold) cells, which is consistent with slower growth kinetics observed for wt MLB1 virus (Fig 3D and 3E).

To assess the replicon activity for MLB1 and MLB2 deletion mutants, all mutations (Fig 1D) were transferred into corresponding RLuc replicon systems and activity was measured during the early (4 h) and later (20, 24, and 30 h) replication stages. A consistent 50% to 150% increase in replicon activity was observed for MLB1R-RLuc-Δ30 and MLB1R-RLuc-Δ74$^{OF}$

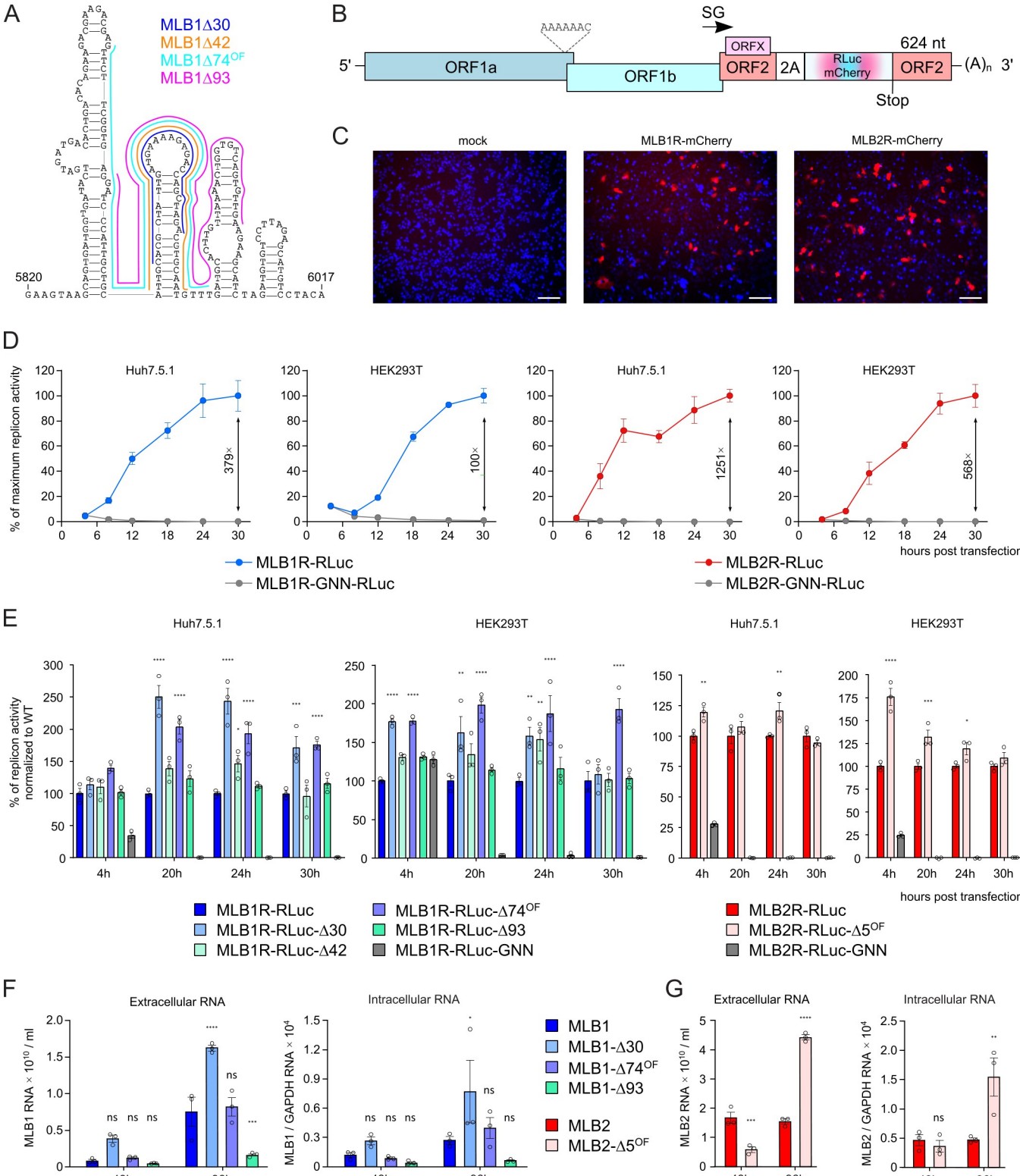

**Fig 5. RNA replication properties in truncated MLB replicons and viruses. (A)** The prediction of the RNA secondary structure and location of identified deletions in MLB1 3′ UTR. **(B)** Schematic of the MLB1 and MLB2 astrovirus replicons. The 2A-RLuc cassette is fused in the ORF2 followed by the stop codon and extended 3′ UTR. **(C)** Huh7.5.1 cells were transfected with MLB1 and MLB2 replicons expressing mCherry, incubated for 24 h, fixed and imaged, nuclei were counterstained with Hoechst (blue). Scale bars are 100 μm. **(D)** Relative MLB1 and MLB2 replicon luciferase activities were measured after RNA

transfection of Huh7.5.1 or HEK293T cells. Values are normalized so that the mean wt replicon value at each time point is 100%. The replication fold difference between wt and GNN mutant replicon is provided for the final time point. **(E)** Relative MLB1 and MLB2 replicon luciferase activities were measured after RNA transfection of Huh7.5.1 or HEK293T cells. The resulting replicon activities were normalized to wt replicon values at each time point. **(F, G)** Extracellular and intracellular RNA levels were measured for MLB1 (F) and MLB2 (G) infections. Huh7.5.1 cells were infected with indicated recombinant viruses of MLB1 and MLB2 at an MOI 0.5 in triplicate, the virus was collected at indicated times post infection, and RNA levels were measured for both extracellular and intracellular fractions. These samples were collected in parallel with those shown in Fig 4D to match the observations. Data are mean ± SEM ($n$ = 3, ≥3 independent experiments, graphs D–G). *$p < 0.05$, **$p < 0.01$, ***$p < 0.001$, ****$p < 0.0001$ using two-way ANOVA test against wt replicon/ infection (E–G). All individual quantitative observations that underlie the data can be found in S1 Data file. MOI, multiplicity of infection; ORF, open reading frame; UTR, untranslated region; wt, wild-type.

mutants when compared to corresponding wt replicons in Huh7.5.1 cells ($p < 0.001$ for 20, 24, and 30 h). In HEK293T cells, MLB1R-RLuc-Δ74$^{OF}$ was replicating above wt levels ($p < 0.0001$ for all time points) with other mutants having nonsignificant or less pronounced effects. A wt-like level was detected for MLB1R-RLuc-Δ42, MLB1R-RLuc-Δ93, and MLB2R-RLuc-Δ5$^{OF}$ replicons at most time points (Fig 5E). These results suggest a beneficial or dispensable nature of identified 3′ RNA structures in the context of replicon system.

Intrigued by these findings, we examined RNA levels in virus-infected Huh7.5.1 cells as well as in media-derived samples. This resulted in increased RNA levels for MLB1-Δ30, wt-like RNA levels for MLB1-Δ74$^{OF}$ and reduced RNA levels for MLB1-Δ93 (Fig 5F). RNA levels in MLB2-Δ5$^{OF}$-infected cells were decreased during initial infection, followed by an increase in all samples at the late infection stage (Fig 5G), coinciding with higher cytotoxicity levels (S1 Fig). Taken together, our results indicate a role of 3′-located elements in the regulation of viral RNA synthesis.

## MLB viruses with 3′ deletions are attenuated in iPSC-derived neurons

To assess the neurotropism of MLB viruses, we infected differentiated i³Neurons with wt and mutant recombinant MLB viruses at MOI 0.5 (Fig 6A). The infection with wt MLB2 reached 40.8% efficiency, whereas MLB2-Δ5$^{OF}$ showed weak signs of infection (4.8%), limited to single infected cells and not showing the spread of infection (Fig 6B and 6D). The infection with wt MLB1 resulted in non-uniform distribution of infected cells with overall 8.6% CP-positive neurons. Conversely, the cells infected with MLB1-Δ30, MLB1-Δ74$^{OF}$, and MLB1-Δ93 resulted in a decrease in infection efficiency (6.9%, 4.8%, and 1.1% CP-positive cells, respectively) with MLB1-Δ93 being the most attenuated in primary neurons (Fig 6C and 6D).

Next, we examined the infectivity of the particles released from the infected neurons by titration on susceptible Huh7.5.1 cells. The analysis of virus release demonstrated that all viruses with deletions have significantly reduced production of infectious particles confirming that MLB astroviruses with deletions are strongly attenuated in neurons and result in decreased infectious virus release. Furthermore, only wt viruses could productively infect neurons on passaging (Fig 6E and 6F), whereas infection with truncated MLB viruses resulted in undetectable virus levels after 3 consecutive passages (Fig 6E). Cytotoxicity in neurons was assessed for both wt and mutant viruses and confirmed the reduced toxicity for all mutant virus infections (S1 Fig). Sequencing of MLB1 and MLB2 viruses passaged in neurons did not reveal any changes throughout the genome, in contrast to deletion- and mutation-prone MLB1 in Huh7.5.1 cells (Table 1).

Next, we measured the levels of intracellular virus RNA in infected neurons. Consistent with lower infectivity (Fig 6B, 6D, and 6E), the virus-specific RNA transcripts were significantly reduced in MLB2-Δ5$^{OF}$ (Fig 6G). These results were also in agreement with the Huh7.5.1 data at early stages of infection (Fig 5G), where cytotoxicity was also low (S1 Fig). Surprisingly, the intracellular virus RNA levels were increased in all MLB1 deletion mutants

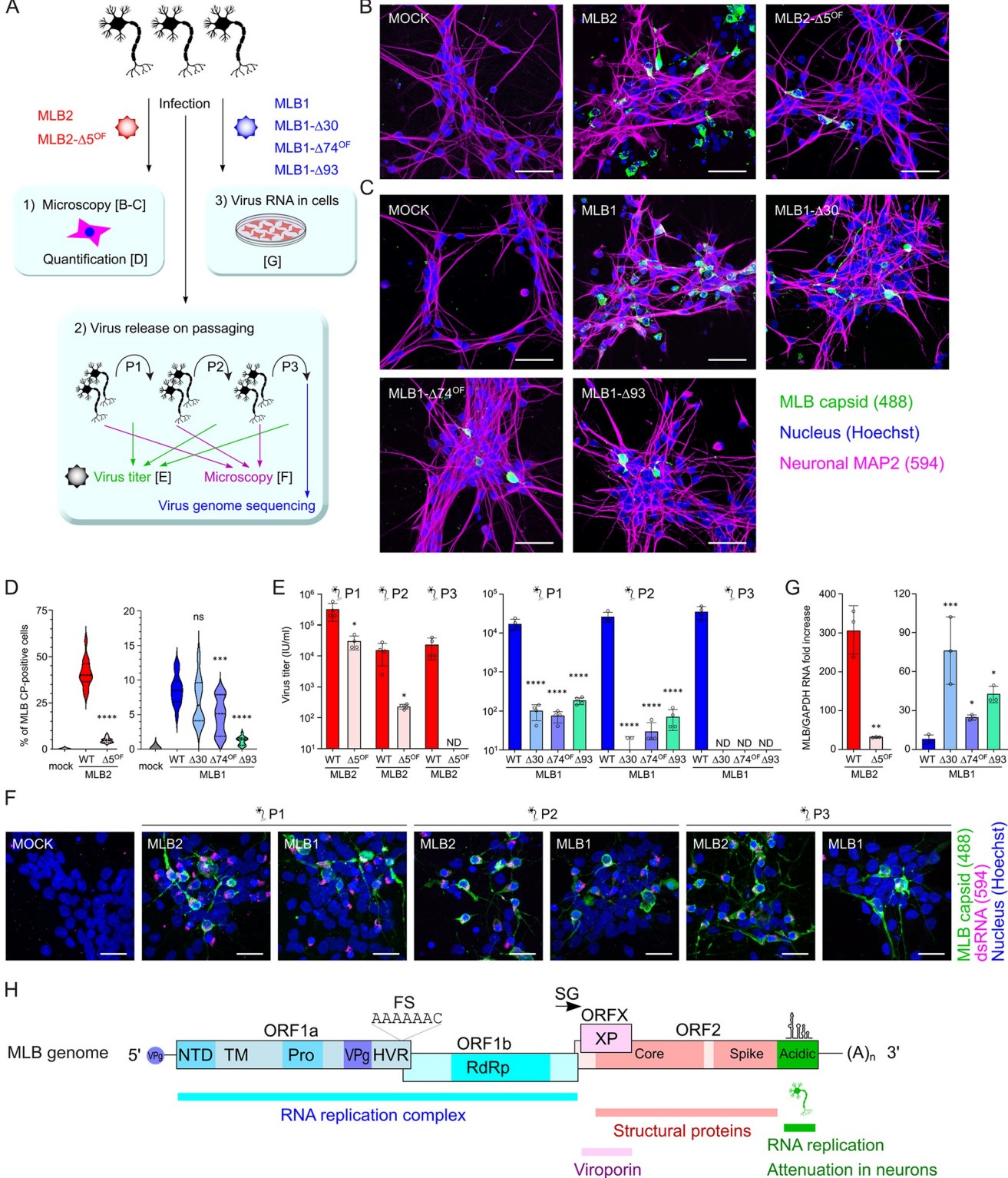

**Fig 6. Infection of iPSC-derived i³Neurons with MLB astroviruses. (A)** The schematic representation of the experiment where partially differentiated i³Neurons were seeded on IBIDI (imaging) or 12-well (other analyses) plates, differentiated into mature i³Neurons, infected with indicated MLB1 and MLB2 recombinant viruses at MOI 0.5. **(B, C)** iPSC-derived neurons were infected with MLB2 (B) and MLB1 (C). Representative confocal images of fixed and permeabilized cells visualized for MLB CP (green) and MAP2 (magenta). Nuclei were stained with Hoechst (blue). Scale bars are 50 μm. **(D)** At least 15 images (150–250 cells per image) were analyzed for CP-positive cells. ***$p < 0.001$, ****$p < 0.0001$, ns nonsignificant, using two-tailed Mann–Whitney

test against wt virus infection. **(E)** iPSC-derived neurons were infected with MLB viruses in quadruplicate at MOI 0.5 and passaged in neurons using 1/100th volume of the previous passage. Each virus was titrated on Huh7.5.1 cells. $*p < 0.05$, $****p < 0.0001$, using two-way ANOVA test against wt infection. **(F)** Representative confocal images of fixed and permeabilized infected i³Neurons were taken, scale bars are 25 μm. **(G)** Intracellular RNA levels were quantified using qPCR. The virus-specific signal was normalized to GAPDH RNA and calculated as fold increase to the input RNA levels. $*p < 0.05$, $***p < 0.001$, using one-way ANOVA test against wt infection (MLB1) and $**p < 0.01$, using Student's $t$ test (MLB2). Data are mean ± SD (D, E, G). **(H)** Schematic representation of MLB genome organization. MLB genome elements: ORF, open reading frame; NTD, N-terminal domain; TM, transmembrane domain; Pro, protease; VPg, virus protein genome-linked; HVR, hypervariable region; RdRp, RNA-dependent RNA polymerase; XP, X protein; FS, frameshift site; SG, subgenomic promoter. All individual quantitative observations that underlie the data can be found in S1 Data file. CP, capsid polyprotein; iPSC, induced pluripotent stem cell; MOI, multiplicity of infection; wt, wild-type.

(Fig 6G) despite lower CP-positive cells and reduced virus release (Fig 6C–6E). Coupled with similar observations in the replicon system (Fig 5E) and in infected Huh7.5.1 cells (Fig 5F), this suggests that RNA replication is strongly unbalanced in MLB1 deletion mutants and may lead to decreased infectivity and particle release in infected neurons. These findings indicate that the 3′ region of MLB1 and MLB2 genome is required for establishing efficient infection in the neuronal cells; however, this is likely achieved in different ways for MLB1 and MLB2 mutants.

Taken together, we developed a powerful platform to investigate neurotropic properties of MLB1 and MLB2 astroviruses and confirmed attenuation of the panel of recombinant viruses with specific 3′ deletions.

## Discussion

A novel MLB group of human astroviruses has gained increasing attention because of invading the non-gastrointestinal tract [4,34–36] and their zoonotic potential [1,14]. However, the basic molecular biology of these viruses is still in its infancy because of the absence of genetically tractable *in vitro* infection models. Here, we report a robust RG system for MLB1 and MLB2 genotypes of human astroviruses that allows the generation of recombinant viruses. We have used this to develop tools to efficiently propagate, visualize, and quantify MLB-infected cells, and utilized this system to characterize MLB genomes with deletions in the 3′ region. Unlike previously reported astrovirus RG systems, MLB RG requires only 1 cell line for both rescue and virus passaging, enabling interrogation of early infection processes such as virus entry, uncoating, and replication. This is particularly important for mutation-prone virus strains when prolonged infection results in substantial changes in the virus genome. As we observed for 2 recombinant MLB astroviruses, one of them (MLB2) had remarkable stability and no changes were detected after 10 serial passages. In contrast, MLB1 was less stable and had more variations, consistent with greater variability of previously reported sequences. This observation, coupled with the developed RG system highlights how investigating MLB represents an opportunity to study differential genomic stability in closely related viruses. Interestingly, clinically isolated MLB2 eventually resulted in 1 deletion upon passaging (MLB2-Δ5^OF), which can be attributed to the quasispecies-nature of clinically isolated viruses.

The processing of CP, virion assembly and maturation in MLB viruses is regulated by several cellular proteases, such as caspases. What is the role of C-terminal deletions in the context of ORF2-encoded polyprotein? The removal of the significant part of the acidic CP portion could affect CP localization, trafficking, particle formation, and potential pro-viral roles that are yet to be characterized for astroviruses. In infected cells, we observed a major large polyprotein precursor and several smaller products (Figs 2C and 3F), suggesting intracellular cleavage of CP. The analysis of media-derived samples revealed a prevalence of smaller CP cleavage products of about 55 kDa (Fig 3G), indicating that CP is cleaved by additional unknown intracellular and/or extracellular proteases. The C-terminal cleavage of CP in MLB2 is likely to be

regulated by cellular caspases (Fig 4F), similar to classical human astroviruses [9]. Despite the presence of the predicted cleavage sites (Fig 4C), the processing of MLB1 capsid is not sensitive to cellular caspase inhibitors (Fig 4F), resembling another neurotropic astrovirus, VA1 [37]. In contrast to neurotropic VA1 and MLB groups of astroviruses, the infectivity of classical astroviruses strongly depends on exogenous trypsin activation [2,37]. Since trypsin is primarily a gut-specific enzyme, this may also explain the extra-gastrointestinal tract tropism of MLB astroviruses, including their ability to infect cell lines of different origins, such as Huh7 (hepatocarcinoma) and A549 (lung adenocarcinoma) [2]. However, it should be noted that brain tissues contain several trypsin-like proteases [38] that may also support the replication of trypsin-dependent viruses [39,40]. Taken together, the requirements for the proteolytic maturation of astrovirus CP represent a powerful strategy to control virus entry, infectivity, and virion maturation, with a likely impact on cellular tropism and pathogenesis.

The deletions identified in the 3′ part of the MLB genomes can result in dual functional defect due to the overlap of structured RNA elements and ORF2 coding sequence. MLB1 and MLB2 have multiple differences in the key properties related to the functionality of the 3′ part of the genome: (i) RNA stem-loop structure was predicted for MLB1 (Fig 5A), but not for MLB2; (ii) sensitivity to caspase inhibition was observed for MLB2, but not MLB1 (Fig 4E); (iii) increased RNA replication in neurons and more significant increase of replicon activity was a hallmark for MLB1 deletion mutants (Figs 5E and 6G). It was logical to expect that the 3′ attenuation in these 2 closely related MLB viruses could have differences in associated RNA and protein-related effects. All deletion mutants identified in MLB1 were mapped to the predicted structured stem-loops that could be beneficial for RdRp processivity in susceptible cells supporting active RNA replication. Enhanced genome replication was indeed observed for MLB1R-RLuc-Δ30 and -Δ74$^{OF}$ replicons (Fig 5E), MLB1-Δ30 in infected Huh7.5.1 cells (Fig 5F), and MLB1-Δ30, -Δ74$^{OF}$ and -Δ93 viruses during infection in neurons (Fig 6G), highlighting the importance of this region in the replication of MLB1 genome. The enhanced replication could have led to the imbalance of RNA replication-translation-packaging and resulted in lower infectious virus particle release and infectivity in neurons (Fig 6C–6E). In contrast, attenuation of MLB2 virus with a small 5-nucleotide out-of-frame deletion resulted in modest differences in replicon activity (Fig 5E), varying RNA levels depending on infection stage in Huh7.5.1 cells (Fig 5G), and 10-fold decreased RNA levels in neuronal infection (Fig 6G), suggesting that differences in cleaved C-terminal part of CP (Fig 1D) could play a major role in attenuation of MLB2. This, however, does not exclude associated RNA defects (Figs 5E and 5G and 6G) due to the overlapping nature of functional elements and tight association between RNA replication, translation, packaging, and virus–host interactions. Observing an imbalanced virus RNA synthesis in several systems and different cell types, we identified a region in the 3′ end of the MLB genome that regulates RNA replication (Fig 6H).

The selection for deletion-prone viruses through serial passaging in highly susceptible cells is a well-known strategy for the generation of vaccine candidates [26]. The viruses with deletions are usually more capable in cell culture and replicate to higher titers, but are highly attenuated in natural hosts, which makes them ideal candidates for live vaccines. Consistently, the deletion of the large portion of the acidic region in the context of the MLB genome could be well tolerated in a highly susceptible Huh7.5.1 cell line but have a distinct role in the terminally differentiated neurons. To our knowledge, this is the first demonstration of attenuation strategy for neurotropic astroviruses, paving the way to the characterization of attenuation mechanisms and elucidation of the corresponding cell- and host-specific pathways. Similar vaccine candidates with deletions in accessory genome regions have been developed for various pathogenic viruses, including delta-6K and delta-5 (nsP3) mutants in Chikungunya virus [41], 30 nucleotide 3′ UTR deletion in Dengue virus [42], SARS-CoV-2 deletions in the multi-basic

cleavage sites of spike protein [43], or combination of several known attenuation approaches that can provide a safer strategy to prevent reversion to virulence [44]. Besides attenuation *in vivo*, trade-offs for enhanced cell culture replication may result in reduced particle stability [45]. Taken together, the identification of the attenuation region in the MLB group of astroviruses brings us a step forward in understanding the mechanisms responsible for virulence in this group of viruses.

There is no animal model for human astroviruses reported so far to elucidate the attenuation mechanism in the context of systemic infection, but it would be interesting to introduce similar deletions in the genomes of closely related neuropathogenic animal astroviruses (Fig 1A) to develop vaccine candidates against circulating astrovirus strains that cause outbreaks in animal farms [17,46,47].

In summary, we have developed a new RG system for MLB astroviruses and have exploited it to identify elements in the 3′ end of the genome that are dispensable for the replication in cell culture but attenuated in human neurons (Fig 6H), thus improving our understanding of the molecular biology of MLB-group astroviruses and facilitating the development of therapeutics and vaccines. The identification of the regions responsible for the neurotropism in the MLB group of astroviruses represents first insights into molecular determinants of neuropathology of astroviruses, advances our understanding in mechanisms involved in this process and help identify virus strains with similar properties. This attenuation strategy can be further applied to other pathogenic human and animal astroviruses.

## Materials and methods

### Cells

HEK293T cells (ATCC) were maintained at 37°C in DMEM supplemented with 10% fetal bovine serum (FBS), 1 mM L-glutamine, and antibiotics. Huh7.5.1 (obtained from Apath, Brooklyn, New York, United States of America) [48] and Caco2 (ATCC) cells were maintained in the same media supplemented with non-essential amino acids (NEAA). All cells were tested mycoplasma negative throughout the work (MycoAlert Mycoplasma Detection Kit, Lonza).

### Plasmids

For bacterial expression of $CNP_{NTD}$, the relevant MLB1 CP-coding sequence corresponding to 61–396 aa in ORF2 was PCR amplified and inserted into the T7 promoter-based pExp-MBP-TEV-CHis expression plasmid with an N-terminal MBP fusion tag, followed by TEV protease cleavage site and C-terminal 8×His-tag (Fig 1A).

To create RG clones for MLB1 and MLB2, the 5′ and 3′ terminal consensus sequences [2] were used to design specific primers to amplify MLB1 and MLB2 full-length genomes using Phusion High-Fidelity DNA polymerase (Thermo Fisher Scientific). The amplified genomes of MLB1 and MLB2 were cloned into the T7 promoter-containing plasmid [20] using a single-step ligation independent cloning. Each 20 μl reaction was prepared using 2 PCR amplicons containing 15 to 20 nucleotide-long overlapping sequences mixed in equimolar proportions (50 ng for shorter product), 1× Buffer 2.1 (NEB), 2 μg BSA, and 3 units of T4 DNA polymerase (NEB) and incubated at 20°C for 30 min. The reaction was stopped by adding 1 μl of 20 mM dGTP, heated to 50°C for 2.5 min, cooled down to room temperature for 20 min, and used for the transformation of XL1 blue competent cells (Agilent). All identified mutations (Fig 5A) were introduced in the MLB1 and MLB2 RG clones using site-directed mutagenesis.

To create MLB1 and MLB2 replicon systems, both MLB RG clones were left intact up to the end of ORFX followed by a 2A sequence and a RLuc or mCherry sequence with a stop codon, followed by the last 624 nt of the virus genome and a 35 nt poly-A tail (Fig 5B). All mutations

were introduced into corresponding replicon plasmids using available restriction sites. The GenBank accession numbers for the pMLB1 and pMLB2 are ON398705 and ON398706, respectively.

All obtained plasmids were sequenced and annotated. The resulting RG and replicon plasmids were linearized with *Xho*I restriction enzyme prior to T7 transcription.

## Purification of His-tagged CP$_{NTD}$ and generation of CP-specific antibody

The MLB1 CP$_{NTD}$ protein was produced in Rosetta 2 (DE3) cells (Novagen) cultured in 2×YT media with overnight expression at 18˚C induced with 0.4 mM IPTG. The protein was purified first by immobilized metal affinity chromatography using PureCube Ni-NTA resin and then by affinity chromatography using amylose resin (NEB). N-terminal MBP fusion tag was removed by the cleavage with TEV protease (produced in-house). The MLB1 CP$_{NTD}$ protein was further purified by heparin chromatography using HiTrap Heparin HP 5 ml column (Cytiva) and, finally, by size exclusion chromatography using a Superdex 200 16/600 column (Cytiva). Protein solution in 50 mM Na-phosphate (pH 7.4), 300 mM NaCl, 5% glycerol was concentrated to 2 mg/ml and used for immunization.

Antibody against CP$_{NTD}$ was generated in rabbits using 5-dose 88-day immunization protocol. Sera were used for CP$_{NTD}$-specific affinity purification, followed by purification of specific IgG fractions (BioServUK).

## Recovery of MLB1 and MLB2 viruses from T7 RNAs

The linearized RG plasmids were used as templates to produce capped T7 RNA transcripts using T7 mMESSAGE mMACHINE T7 Transcription kit (Invitrogen) according to the manufacturer's instructions. For a virus recovery, $10^7$ Huh7.5.1 cells were trypsinized, washed with PBS, and electroporated with 20 μg T7 RNA in 800 μl PBS pulsed twice at 800 V and 25 μF using a Bio-Rad Gene Pulser Xcell electroporation system. The cell suspension was supplemented with 10% FBS-containing media and incubated at 37˚C. After 3 h of incubation and full cell attachment, the media was replaced with serum free media, and cells were incubated until appearance of CPE. To produce recombinant MLB1 virus stocks, electroporated cells were incubated for 72 to 96 h, freeze–thawed twice, filtered through 0.2 μm filter, supplemented with 5% glycerol and stored in small aliquots at −70˚C. For recombinant MLB2 virus stocks, electroporated cells were incubated for 48 to 72 h, the supernatant was clarified by filtration through 0.2 μm filter, supplemented with 5% glycerol, and stored in small aliquots at −70˚C.

## Virus passaging, growth curves, and titration

To passage recombinant MLB1 virus stocks, Huh7.5.1 cells were infected at an MOI 0.1 for 2 h in serum-free media, then 5% FBS-containing media was added and incubated for 16 to 24 h, then replaced with serum-free media and incubated for 72 to 96 h until the appearance of CPE, freeze–thawed twice, filtered through 0.2 μm filter, and supplemented with 5% glycerol. To passage recombinant MLB2 virus stocks, Huh7.5.1 cells were infected at an MOI 0.1 for 2 h in serum-free media, then 5% FBS-containing media was added and incubated for 16 to 24 h, then replaced with serum-free media and incubated for 48 to 72 h until the appearance of CPE; the supernatant was clarified by filtration through 0.2 μm filter and supplemented with 5% glycerol. Virus RNA was isolated by Direct-zol RNA MicroPrep (Zymo research), followed by RT-PCR and Sanger sequencing of the virus genome. The genome ends were sequenced using 5′/3′ RACE kit (Roche).

To concentrate media-derived samples, the infected Huh7.5.1 cells were incubated for 96 h, the media was collected, clarified using 0.2 μm filter, and pelleted at 54,000 rpm (180,000 ×g) for 2 h at 4°C in a TLA-55 rotor in an Optima Max-XP tabletop ultracentrifuge (Beckman). The supernatant was removed, the pellet was resuspended in 50 mM Tris (pH 6.8) to obtain 20× concentrated sample and analyzed by SDS-PAGE followed by western blotting.

Multistep growth curves were performed using an MOI of 0.1 (Fig 3D and 3E) or 0.5 (Fig 4D). Individual infections were performed in triplicates. Both cell- and media-derived samples were collected in equal volume at indicated times post infection and saved for virus quantification.

Virus competition assay was performed in duplicates using equal amounts of wt and mutant viruses at MOI 0.1. The passaging was performed using 1/1,000 volume of the previous passage. RNA was extracted using the Qiagen QIAamp viral RNA mini kit. Reverse transcription was performed using the Superscript III reverse transcriptase (Thermo Fisher Scientific) as per manufacturer's protocol. The PCR primers were designed to amplify 350(wt)/320(Δ30)/276(Δ74$^{OF}$)/257(Δ93) bp fragments for MLB1 and 60(wt)/55(Δ5$^{OF}$) bp fragments for MLB2.

The immunofluorescence-based detection with anti-CP$_{NTD}$ MLB1 antibody (1:300) was combined with infrared detection readout and automated LI-COR software-based quantification. Briefly, 24 h before infection, Huh7.5.1 cells were plated into 96-well plates (2–3×10$^4$ cells/well). The 10-fold serial dilutions of virus stock in a round-bottom 96-well plate were prepared using serum-free media supplemented with 0.2% BSA. The cells were infected with prepared virus dilutions in duplicates, incubated for 24 h, fixed with 4% paraformaldehyde (PFA), processed for immunofluorescence staining, scanned and counted as the number of capsid-positive signals. The titers were determined as infectious units per ml (IU/ml).

## SDS-PAGE and immunoblotting

Protein samples were analyzed using 8% SDS-PAGE. The resolved proteins were then transferred to 0.2 μm nitrocellulose membranes and blocked with 4% Marvel milk powder in PBS. Immunoblotting of MBL1 and MLB2 capsid proteins was performed using anti-CP$_{NTD}$ MLB1 antibody (custom-made rabbit polyclonal antibody, 1:3,000). Anti-tubulin antibody (Abcam, ab6160, 1:1,000) was used for the cellular target. Secondary antibodies (LI-COR IRDye 800 and 680, 1:3,000) were used for IR-based detection. Immunoblots were imaged on a LI-COR ODYSSEY CLx imager and analyzed using Image Studio version 5.2.

## Inhibition of caspase cleavage in astrovirus-infected cells

Caco2 cells were infected with HAstV4, Huh7.5.1 cells were infected with MLB1 and MLB2 astroviruses (MOI 1) in the presence or absence of 20 μm z-VAD-fmk (pan-caspase inhibitor, Promega). At indicated time post infection, cells were lysed and analyzed by immunoblotting using virus-specific antibodies. The CP of HAstV4 was detected using astrovirus 8E7 antibody (Santa Cruz Biotechnology, sc-53559, 1:750).

## Analysis of CPE and fluorescent microscopy

For the analysis of virus-induced CPE, plasma membranes of infected cells were stained with Wheat Germ Agglutinin Alexa Fluor 488 Conjugate (WGA, Thermo Fisher Scientific, 1:250) for 20 min, followed by fixation with 4% PFA for 20 min, permeabilization with 0.05% Triton X-100 in PBS for 15 min, and nuclei counter-staining with Hoechst (Thermo Fisher Scientific) for 15 min. Cells were washed twice with PBS and imaged using EVOS fluorescence microscope. For the analysis of CP localization, the infected Huh7.5.1 cells were incubated for 24 h, fixed and permeabilized as above. CP was detected using anti-CP$_{NTD}$ MLB1 antibody followed

by incubation with secondary antibody (Alexa Fluor 488-conjugated goat anti-rabbit, Thermo Fisher, A21441). Nuclei were counter-stained with Hoechst. The images are single plane images taken with a Leica SP5 Confocal Microscope using a water-immersion 63× objective.

## MLB replicon assay

Linearized replicon-encoding plasmids were utilized to produce T7 RNAs using mMESSAGE mMACHINE T7 Transcription kit, purified using Zymo RNA Clean & Concentrator kit and quantified. Huh7.5.1 and HEK293T cells were transfected in triplicate with Lipofectamine 2000 reagent (Invitrogen), using the reverse transfection protocol. Briefly, a mixture of 0.5 μl Lipofectamine 2000 and 0.5 μl OMRO (OptiMEM containing 40 units/ml RNaseOUT) was incubated for 5 min at room temperature before adding to a mixture of 100 ng T7 replicon RNA, 10 ng T7 Firefly luciferase-encoding RNA, and 10 μl OMRO per transfection. After 20 min incubation at room temperature, 100 μl of the prewashed cells ($10^5$ cells) were added to the transfection mixture, incubated at room temperature for 5 min, supplemented with 5% FBS, transferred to a 96-well plate, and incubated for indicated time at 37°C (4 to 30 h). Replicon activity was calculated as the ratio of Renilla (subgenomic reporter) to Firefly (co-transfected loading control RNA, cap-dependent translation) using Dual Luciferase Stop & Glo Reporter Assay System (Promega) and normalized by the same ratio for the control wt replicon. Three independent experiments, each in triplicate, were performed to confirm the reproducibility of the results. To control for the transfection efficiency, MLB1 and MLB2 replicons encoding mCherry fluorescent protein were also transfected and visualized using EVOS fluorescent microscope (Thermo Fisher Scientific).

## Growth and infection of iPSC-derived i³Neurons

i³Neuron stem cells were maintained at 37°C in complete E8 medium (Gibco) on plates coated with Matrigel (Corning) diluted 1:50 in DMEM. Initial three-day differentiation was induced with DMEM/F-12, HEPES (Gibco) supplemented with 1× N2 supplement (Thermo), 1× NEAA, 1× Glutamax, and 2 μg/ml doxycycline. Approximately 10 μm Rock Inhibitor (Y-27632, Tocris) was added during initial plating of cells to be differentiated and cells were plated onto Matrigel-coated plates. Differentiation medium was replaced daily and after 3 days of differentiation, partially differentiated neurons were re-plated into Cortical Neuron (CN) media in IBIDI wells coated with 100 μg/ml poly-L-ornithine (Sigma). CN media consisted of: Neurobasal Plus medium (Gibco) supplemented with 1× B27 supplement (Gibco), 10 ng/ml BDNF (Peprotech), 10 ng/ml NT-3 (Peptrotech), 1 μg/ml laminin (Gibco), and 1 μg/ml doxycycline. After initial replating, neurons were then maintained in CN media without doxycycline for a further 11 days until mature.

The fully differentiated neurons were infected with MLB astroviruses at an MOI 0.5 in neuronal media. After 2 h, the virus inoculum was removed and replaced with 50% fresh– 50% conditioned media. After 48 hpi (MLB2) or 96 hpi (MLB1), the cells grown on IBIDI wells were fixed, permeabilized, and stained with anti-CP$_{NTD}$ MLB1 antibody followed by incubation with Alexa 488-conjugated secondary antibody, followed by the staining with antibody against neuronal marker MAP2 (Abcam, ab11268) or anti-dsRNA IgG2a monoclonal antibody (Scicons J2, 10010500) and Alexa 594-conjugated secondary antibody. Nuclei were counter-stained with Hoechst. The confocal images are a projection of a z-stack images taken with a Leica SP5 Confocal Microscope using a water-immersion 63× objective. To analyze the percentage of CP-positive cells, the images were taken with EVOS fluorescence microscope. At least 15 images (150 to 250 cells per image) were analyzed. The infectivity of the particles released from the infected neurons was determined by titration on susceptible Huh7.5.1 cells

as described above. Reinfection experiments were performed using 1/100 volumes of the media obtained from the previous passage.

## Analysis of RNA levels in samples collected from infected cells and virus stocks

Terminally differentiated neurons grown on 12-well plates (400,000 cells per well) were infected at MOI 0.5 and collected at 72 hpi. Huh7.5.1 cells grown on 12-well plates (400,000 cells per well) were infected at MOI 0.5 and collected at indicated hpi. RNA was isolated by Direct-zol RNA MicroPrep (Zymo research), followed by quantitative reverse transcription-PCR (RT-qPCR) detection of virus (MLB1, MLB2) and cell-specific (GAPDH) RNAs. Results were normalized to the amount of GAPDH RNA in the same sample. Fold differences in RNA concentration were calculated using the $2^{-\Delta\Delta CT}$ method.

The absolute amount of MLB RNA in virus stock was determined by RT-qPCR. A 20 μl aliquot of each sample was mixed with $4 \times 10^6$ plaque-forming units of purified Sindbis virus (SINV) stock, which was used to control the quality of RNA isolation. RNA was extracted using the Qiagen QIAamp viral RNA mini kit. Reverse transcription was performed using the QuantiTect reverse transcription kit (Qiagen) using virus-specific reverse primers for SINV (GTTGAAGAATCCGCATTGCATGG), MLB1 (GTTGCACTGGCACCAGAGTC), MLB2 (GTGATAGTGAGGGATCTTCTGC). The known genome copy MLB standards were prepared using quantified purified T7 RNA transcripts of full-length MLB genomes.

Quantitative PCR was performed in triplicate using SsoFast EvaGreen Supermix (Bio-Rad) in a ViiA 7 Real-time PCR system (Applied Biosystems) for 40 cycles with 2 steps per cycle. MLB and SINV-specific primers were used to quantify corresponding virus RNAs; the primer efficiency was within 95% to 105%.

qPCR primers: SINV-F (GAAACAATAGGAGTGATAGGCA), SINV-R (TGCATACC CCTCAGTCTTAGC), GAPDH-F (GCAAATTCCATGGCACCGT), GAPDH-R (TCGC CCCACTTGATTTTGG), MLB1-F (TTGCCAAGTGAGCCTTACAAAC), MLB1-R (TGC CATCAACAACTGGAAGCAC), MLB2-F (GATGTCTTTGGAATGTGGGTAAAG), MLB2-R (CTAGGTGCAGGTCCTTTCTTAG).

## Cytotoxicity analysis

Terminally differentiated neurons and Huh7.5.1 cells grown on 96-well plates (100,000 cells per well) were infected at MOI 0.5 in quintuplicate. Cytotoxicity was assessed by live-cell imaging using an Incucyte SX5, an automated phase-contrast and fluorescence microscope within a humidified incubator, using Cytotox NIR Dye (Sartorius) at 1:1,000 dilution added directly during virus infection. At 4-hour intervals, 3 images per well were taken and used to estimate NIR-positive cells per well using the integrated software. For each experiment, the data were plotted as NIR object count per image normalized to the 0 hpi time point.

## Analysis of MLB1 and MLB2 sequences using the NCBI database

Available MLB1 and MLB2 complete and ORF2-specific genome sequences from NCBI deposited before January 2022 were identified and extracted from NCBI using Blastp. These sequences were then aligned using MUSCLE (https://www.ebi.ac.uk/Tools/msa/muscle/) and visualized using Bioedit software.

### RNAfold analysis of 3′ terminal MLB sequences

The secondary structures of the 3′ terminal MLB1 and MLB2 sequences were predicted by the RNAfold Server using default settings [49].

### Prediction of caspase putative cleavage sites

The putative caspase cleavage sites in C-terminal part of the MLB1 and MLB2 CP were analyzed by Procleave software [33] using default settings. Only predicted cleavage sites with probability score of >0.7 were considered.

### Statistical analyses

Data were graphed and analyzed using GraphPad Prism and MS Excel. Where appropriate, data were analyzed using one-way, two-way ANOVA, Student's $t$ test or two-tailed Mann–Whitney test. Significance values are shown as $****p < 0.0001$, $***p < 0.001$, $**p < 0.01$, $*p < 0.05$, ns–nonsignificant.

## Supporting information

**S1 Fig. Cytotoxicity assay.** Terminally differentiated neurons and Huh7.5.1 cells grown on 96-well plates were infected at MOI 0.5 in quintuplicates. Cytotoxicity was assessed by live-cell imaging using Cytotox NIR Dye added directly during virus infection. **(A)** NIR object counts data for Huh7.5.1 cells infected with MLB1 and MLB2, normalized to 0 hpi. **(B)** Representative images of MLB-infected Huh7.5.1 cells at 96 hpi. **(C)** Cytotoxicity analysis of MLB1 and mutant infected Huh7.5.1 cells, normalized to maximum MLB1 toxicity. **(D)** Cytotoxicity analysis of MLB2 and mutant infected Huh7.5.1 cells, normalized to maximum MLB2 toxicity. **(E)** NIR object counts data for i³Neurons infected with MLB1 and MLB2, normalized to 0 hpi. **(F)** Representative images of MLB-infected i³Neurons at 60 hpi. **(G)** Cytotoxicity analysis of MLB1 and mutant infected i³Neurons, normalized to maximum MLB1 toxicity. **(H)** Cytotoxicity analysis of MLB2 and mutant infected i³Neurons, normalized to maximum MLB2 toxicity. All data are mean ± SEM ($n = 5$), ns, nonsignificant, $*p < 0.05$, $**p < 0.01$, $***p < 0.001$, $****p < 0.0001$ using two-way ANOVA with repeated measures test against wt MLB infection. All individual quantitative observations that underlie the data can be found in S1 Data file. (TIF)

**S1 Data. Quantitative data underlying each figure (Figs 2–6 and S1).** Each tab contains the panels relative to the indicated figure for which a quantitative analysis was required. (XLSX)

**S1 Raw Images. Raw images relative to the blots and gel images included in the manuscript.** (PDF)

## Acknowledgments

Authors thank Andrew Firth and Marko Hyvönen (University of Cambridge) for their support at the beginning of this project. We thank the Cambridge NIHR BRC Cell Phenotyping Hub for access to confocal microscopy.

## Author Contributions

**Conceptualization:** Valeria Lulla.

**Data curation:** Valeria Lulla.

**Formal analysis:** Hashim Ali, Aleksei Lulla, Jacqueline Hankinson, Valeria Lulla.

**Funding acquisition:** Janet E. Deane, Valeria Lulla.

**Investigation:** Hashim Ali, Aleksei Lulla, Alex S. Nicholson, Jacqueline Hankinson, Elizabeth B. Wignall-Fleming, Rhian L. O'Connor, Diem-Lan Vu, Stephen C. Graham, Janet E. Deane, Susana Guix, Valeria Lulla.

**Methodology:** Hashim Ali, Aleksei Lulla, Alex S. Nicholson, Rhian L. O'Connor, Diem-Lan Vu, Stephen C. Graham, Janet E. Deane, Susana Guix, Valeria Lulla.

**Project administration:** Valeria Lulla.

**Resources:** Hashim Ali, Aleksei Lulla, Alex S. Nicholson, Stephen C. Graham, Janet E. Deane, Susana Guix, Valeria Lulla.

**Supervision:** Janet E. Deane, Susana Guix, Valeria Lulla.

**Validation:** Valeria Lulla.

**Visualization:** Hashim Ali, Jacqueline Hankinson, Valeria Lulla.

**Writing – original draft:** Hashim Ali, Valeria Lulla.

**Writing – review & editing:** Hashim Ali, Aleksei Lulla, Alex S. Nicholson, Jacqueline Hankinson, Elizabeth B. Wignall-Fleming, Stephen C. Graham, Janet E. Deane, Susana Guix, Valeria Lulla.

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
