## [Editor Report · Decision Letter 0]

2 Sep 2022

Dear Dr. Lulla, 

Thank you for submitting your manuscript entitled "Attenuation hotspots in neurotropic human astroviruses" for consideration as a Research Article by PLOS Biology.

Your manuscript has now been evaluated by the PLOS Biology editorial staff, as well as by an academic editor with relevant expertise, and I am writing to let you know that we would like to send your submission out for external peer review. In our discussion we thought that your manuscript will work better as a Methods and Resources paper. Please, select this option when re-submitting your manuscript. 

PLOS Biology ‘Methods’ should report a novel method or improvements to current methodologies that significantly outperform the existing state-of-the-art methods or that show the potential to address, for the first time, a pressing biological question. Ideally, these Methods papers should be of broad interest. PLOS Biology Resources should be truly exceptional, in such a way as to spur future research. While we value the inclusion of novel biological insight in our ‘Methods and Resources’ articles, this is not a condition for consideration in PLOS Biology.

Before we can send your manuscript to reviewers, we need you to complete your submission by providing the metadata that is required for full assessment. To this end, please login to Editorial Manager where you will find the paper in the 'Submissions Needing Revisions' folder on your homepage. Please click 'Revise Submission' from the Action Links and complete all additional questions in the submission questionnaire.

Once your full submission is complete, your paper will undergo a series of checks in preparation for peer review. After your manuscript has passed the checks it will be sent out for review. To provide the metadata for your submission, please Login to Editorial Manager (https://www.editorialmanager.com/pbiology) within two working days, i.e. by Sep 04 2022 11:59PM.

Kind regards,

Paula

---

Senior Editor

PLOS Biology

---

## [Decision Letter · Decision Letter 1]

29 Nov 2022

Dear Dr. Lulla,

Please allow me to first apologize for the delay in the processing of your manuscript. This delay is caused by my difficulty in recruiting reviewers for your manuscript. I am sorry for this unexpected event, and I thank you for your patience while your manuscript "Attenuation hotspots in neurotropic human astroviruses" was peer-reviewed at PLOS Biology. Your manuscript has been evaluated by the PLOS Biology editors, an Academic Editor with relevant expertise, and by several independent reviewers. 

As you will see in the reviewer reports, which can be found at the end of this email, although the reviewers find the work potentially interesting, they have also raised a substantial number of important concerns. Based on their specific comments and following discussion with the Academic Editor, it is clear that a substantial amount of work would be required to meet the criteria for publication in PLOS Biology. However, given our and the reviewer interest in your study, we would be open to inviting a comprehensive revision of the study that thoroughly addresses all the reviewers' comments. Given the extent of revision that would be needed, we cannot make a decision about publication until we have seen the revised manuscript and your response to the reviewers' comments. Your revised manuscript would need to be seen by the reviewers again, but please note that we would not engage them unless their main concerns have been addressed. 

Having discussed the reviews with the Academic Editor, we think that it is important for you to demonstrate the neural tropism of the viruses and the utility and faithfulness of the recombinant system. It would be particularly compelling if the results are validated by real world phylogenetic comparisons, if the sequences exist. If not, passaging in neuronal cells and fortifying those experiments would be needed.

We appreciate that these requests represent a great deal of extra work, and we are willing to relax our standard revision time to allow you 6 months to revise your study. Please email us (plosbiology@plos.org) if you have any questions or concerns, or envision needing a (short) extension.

**IMPORTANT - SUBMITTING YOUR REVISION**

*Resubmission Checklist*

*Published Peer Review*

*PLOS Data Policy*

*Blot and Gel Data Policy*

Sincerely,

Paula

---

Senior Editor

PLOS Biology

REVIEWS:

Reviewer #1: Astrovirus pathogenesis.

Reviewer #2: Virus evolution.

Reviewer #3: Astroviruses.

Reviewer #1: In this manuscript, Ali et al undertake an ambitious and comprehensive series of experiments to define regions of the astrovirus MLB genome required for replication in neuronal cells. The authors take a directed evolution approach to adapt clinical MLB isolates and then develop a battery of tools to identify the genomic changes associated with replication in cells including human i3 neurons. Yet, the manuscript primarily focuses on tool development/characterization and using their newly described reverse genetics system to assess genomic stability, analyze naturally occurring mutations in the genomes, and creating recombinant viruses with deletions to assess growth in Huh7.5 cells. There are minimal studies characterizing growth in neuronal cells and no evidence that MLB1 or 2 productively replicates in these cells. This is a major concern about the studies. Further, replication in a single neuronal cell ex vivo fails to address neurotropism. Based on the presented data, the authors are strongly encouraged to refocus the manuscript on characterization of the new reverse genetics system and understanding the areas of the genome involved in replication in Huh7.5 cells, which is the strongest part of the studies. 

Major

1. The majority of the studies focus on understanding adaptations generated in serially passaging MLB1 and MLB2 co-infected Huh7.5 cells. There is no discussion on why co-infections are used. There is no evidence from human studies that MLB1 and 2 co-infect people. It is also unclear why both strains are used rather than focusing on the more frequently isolated MLB1 strain.

2. Is there evidence from comparative phylogenetic studies, that the genomes of CNS-isolated MLB strains differ from those obtained from the intestines? Given that these viruses are a major cause of CNS-associated infections in several animal species, these sequences should be available for analysis.

3. More information is needed on the source of the clinical isolates used and whether they were derived from enteric or CNS infection.

4. A major premise of the study is identifying the regions of the viral genome associated with neurotropism. The author's tackle this by adapting clinical MLB viruses in cell culture and constructing recombinant viruses to study replication in particular neuronal cells in culture. Yet, the adaptation takes place in non-neuronal, non-biologically relevant Huh 7.5 cells. It is unclear how this would translate the in vivo situation. These studies should be repeated in neuronal cells to assess the molecular changes that occur. 

5. The authors fail to demonstrate that MLB viruses productively replicate in neuronal cells in vitro. Further, the references cited do not support their claim that these viruses infect cells of neuronal origin.

Reviewer #2: This article describes a novel reverse genetics system that should help us understand the biology of the MLB genogroup of astroviruses. The main advantage over previous systems is that transfection, recovery, and subsequent viral propagation are performed in the same cell line, as opposed to previous systems using two cell lines (one for recovery and another for propagation). The article is essentially methodological and, although the proposed techniques should be relevant, no major advances in our understanding of these astroviruses is reported here. I thus find it questionable whether this manuscript brings sufficient advance for publication in a generalist journal such as PLoS Biol, or should be preferably published in a more specialized journal in the field of virology. 

Specific comments:

1- To better appreciate the advantages of the reverse genetics system developed here, the authors should further discuss the limitations of the previous strategies. Given that recovery is apparently straightforward (transfection of a T7 in vitro transcript), why the previous systems cannot be implemented in a single cell line?

2- How was the sequencing performed? Specifically, how were the 5´and 3´ genome ends sequenced?

3- lines 113-120: I don´t see how these two viruses could complement if the MOI was 0.1.

4- Why was the recovered virus stable in permissive cells (Huh7.5), whereas the clinical isolates evolved more mutations? Information about how these clinical isolates were passaged is missing.

5- The deletions that emerged in the serially passaged viruses should be beneficial under the conditions of this evolution experiment, unless there is a specific reason for their emergence other than positive selection. Could the authors perform competition assays to test this? This benefit is suggested by the replicon experiments, but could be further shown using full viruses. Interestingly, the mutants seem to exhibit accelerated replication in neurons but an overall fitness cost due to deficient virus gene expression, assembly or release. It would be interesting to also have data on replication kinetics and total viral fitness in the permissive cell line to more precisely identify the stage of the infection cycle where differences between Huh7.5 cells and neurons take place. 

Reviewer #3: The manuscript by Ali et al. provides an exciting new advance to the study of human astroviruses, MLB1 and MLB2, in the form of molecular clones and replicon systems. With these new systems the authors define a difference in genomic stability between the two viruses as well as a difference in reliance on caspase cleavage for capsid production. Furthermore, this study demonstrates the novelty of using replication deficient astroviruses as a means to drive attenuation and the prospect of developing live-attenuated strains for vaccines. Overall the data and experimentation are clear and provide sound conclusions. However, the following suggestions would help clarify a few key points in the manuscript: 

Major 

Revisions of Fig1a and accompanying figure legend should be considered since most, or possibly all, sequences in the tree were obtained from stool samples collected during gastrointestinal illness. Therefore, it is difficult to assign strictly neuronal tropism to all members of those clades. Additionally, 2 classical HAstV cases of neurological disease have been identified (Koukou 2019, Wunderli 2011) but the neuron picture was not included for that clade.

In the competition experiment (Fig 1c), it is noted that there was no evidence of out-competition for 10 consecutive passages. As a point of clarification- were the same levels (ie. titer) of the virus found after each passage or were the titers after each passage not significantly different than when the viruses are passaged separately? 

Fig 6G and the accompanying data convincingly show that the deletion mutants result in much higher ratios of defective particles. Was there evidence of this during the sequencing of passaged viruses?

Line 422: While trypsin is typically thought as being gut-specific, the brain also houses several trypsin-like proteases. Therefore, this explanation for extra-gastrointestinal disease associated with MLB and VA1 strains may be insufficient.

While the attenuation noted in Fig6 indicates reduced infection, the presence of virus particles, including defective particles, may still trigger inflammation and cell death of neurons. Such experiments may be needed in future evaluation of live-attenuated strains for vaccine development. 

Minor 

Can the authors include the passage number that was used to obtain the data in Fig3D and E?

Line 160: was it previously shown that passaging the virus in Huh7.5.1 results in enhanced replication and CPE compared to Huh7?

Line 240: Point of clarification- "…all related WT MLB astrovirus sequences" were those available in GenBank?

---

## [Decision Letter · Decision Letter 2]

19 May 2023

Dear Dr. Lulla,

Thank you for your patience while we considered your revised manuscript "Attenuation hotspots in neurotropic human astroviruses" for publication as a Methods and Resources at PLOS Biology. This revised version of your manuscript has been evaluated by the PLOS Biology editors, the Academic Editor, and the original reviewers.

Based on the reviews and on our Academic Editor's assessment of your revision, we are likely to accept this manuscript for publication, provided you satisfactorily address the remaining points raised by reviewer #2. To address the remaining concerns from reviewer #2, we suggests that you make a more nuanced conclusion explicitly stating that complementation is the likely explanation. It is important that you maintain a focus on describing the data.

Please also make sure to address the following data and other policy-related requests.

1. DATA POLICY:

Regardless of the method selected, please ensure that you provide the individual numerical values that underlie the summary data displayed in the following figure panels as they are essential for readers to assess your analysis and to reproduce it: 2EH, 3DEHI, 4D, 5DEFG, 6DEG, S1ACDEGH (We are aware that you provided the underlying data).

**Please also ensure that figure legends in your manuscript include information on where the underlying data can be found, and ensure your supplemental data file/s has a legend.**

We require the original, uncropped and minimally adjusted images supporting all blot and gel results reported in an article's figures or Supporting Information files. We will require these files before a manuscript can be accepted so please prepare and upload them now. We need this for figure 2B.

Please carefully read our guidelines for how to prepare and upload this data: https://journals.plos.org/plosbiology/s/figures#loc-blot-and-gel-reporting-requirements

3. We suggest a more appropriate title for a Methods and Resources manuscript: "Identification of attenuation hotspots in human neurotropic astroviruses uncovers determinants of neurotropism and provides a platform for attenuated vaccine development". Please feel free to modify as you think fits better.

We expect to receive your revised manuscript within two weeks.

*Published Peer Review History*

*Press*

Sincerely,

Paula

---

Senior Editor,

pjaureguionieva@plos.org,

PLOS Biology

Reviewer remarks:

Reviewer #1: No further concerns. Interesting manuscript bringing important new information to the field.

Reviewer #2: As indicated in my previous review, this paper presents an interesting experimental setup that could help in the development of astrovirus culture and attenuated vaccines, but in my opinion does not report major advances in our understanding of astrovirus biology. For example, encephalitis-causing astroviruses have previously been cultured in neurons. https://www.ncbi.nlm.nih.gov/pmc/articles/PMC6747723/. The fact that serial transfer in a permissive cell line such as Huh7.5 leads to the accumulation of conditionally deleterious mutations is not surprising. Some of these mutations (but potentially many others) could be exploited for the development of live attenuated vaccines.

Regarding my previous point 3, even if the MOI rises above 1 in secondary infection cycles, complementation is wiped out in each transfer. The rationale and interpretation of this passaging strategy remains therefore unclear.

Reviewer #3: All comments have been adequately addressed, thank you and well done!

---

## [Editor Report · Decision Letter 3]

13 Jun 2023

Dear Dr. Lulla,

Thank you for the submission of your revised Methods and Resources "Attenuation hotspots in neurotropic human astroviruses" for publication in PLOS Biology. On behalf of my colleagues and the Academic Editor, Ken Cadwell, I am pleased to say that we can in principle accept your manuscript for publication, provided you address any remaining formatting and reporting issues. These will be detailed in an email you should receive within 2-3 business days from our colleagues in the journal operations team; no action is required from you until then. Please note that we will not be able to formally accept your manuscript and schedule it for publication until you have completed any requested changes.

PRESS

Sincerely, 

Paula

---

Senior Editor

PLOS Biology
